SCIENCE FORUM

# The Brazilian Reproducibility Initiative

**Abstract** Most efforts to estimate the reproducibility of published findings have focused on specific areas of research, even though science is usually assessed and funded on a regional or national basis. Here we describe a project to assess the reproducibility of findings in biomedical science published by researchers based in Brazil. The Brazilian Reproducibility Initiative is a systematic, multicenter effort to repeat between 60 and 100 experiments: the project will focus on a set of common methods, repeating each experiment in three different laboratories from a countrywide network. The results, due in 2021, will allow us to estimate the level of reproducibility of biomedical science in Brazil, and to investigate what aspects of the published literature might help to predict whether a finding is reproducible.
DOI: https://doi.org/10.7554/eLife.41602.001

**OLAVO B AMARAL\*, KLEBER NEVES, ANA P WASILEWSKA-SAMPAIO AND CLARISSA FD CARNEIRO**

**\*For correspondence:** olavo@bioqmed.ufrj.br

**Competing interests:** The authors declare that no competing interests exist.

## Introduction

Concerns about the reproducibility of published results in certain areas of biomedical research were initially raised by theoretical models (*Ioannidis, 2005a*), systematic reviews of the existing literature (*Ioannidis, 2005b*) and alarm calls by the pharmaceutical industry (*Begley and Ellis, 2012*; *Prinz et al., 2011*). These concerns have subsequently been covered both in scientific journals (*Baker, 2016*) and in the wider media (*Economist, 2013*; *Harris, 2017*). While funding agencies have expressed concerns about reproducibility (*Collins and Tabak, 2014*), efforts to replicate published findings in specific areas of research have mostly been conducted by bottom-up collaborations and supported by private funders. The Reproducibility Project: Psychology, which systematically reproduced 100 articles in psychology (*Open Science Collaboration, 2015*), was followed by similar initiatives in the fields of experimental economics (*Camerer et al., 2016*), philosophy (*Cova et al., 2018*) and social sciences (*Camerer et al., 2018*), with replication rates ranging between 36 and 78%. Two projects in cancer biology (both involving the Center for Open Science and Science Exchange) are currently ongoing (*Errington et al., 2014*; *Tan et al., 2015*).

Although such projects are very welcome, they are all limited to specific research topics or communities. Moreover, apart from the projects in cancer biology, most have focused on areas of research in which experiments are relatively inexpensive and straightforward to perform: this means that the reproducibility of many areas of biomedical research has not been studied. Furthermore, although scientific research is mostly funded and evaluated at a regional or national level, the reproducibility of research has not, to our knowledge, been studied at these levels. To begin to address this gap, we have obtained funding from the Serrapilheira Institute, a recently created nonprofit institution, in order to systematically assess the reproducibility of biomedical research in Brazil.

Our aim is to replicate between 60 and 100 experiments from life sciences articles published by researchers based in Brazil, focusing on common methods and performing each experiment at multiple sites within a network of collaborating laboratories in the country. This will allow us to estimate the level of reproducibility of research published by biomedical scientists in Brazil, and to investigate if there are aspects of the published literature that can help to predict whether a finding is reproducible.

## Brazilian science in a nutshell

Scientific research in Brazil started to take an institutional form in the second half of the 20th century, despite the earlier existence of important organizations such as the Brazilian Academy of Sciences (established in 1916) and the Universities of Brazil (later the Federal University of Rio de Janeiro) (1920) and São Paulo (1934). In 1951, the federal government created the first national agency dedicated to funding research (CNPq), as well as a separate agency to oversee postgraduate studies (CAPES), although graduate-level education was not formalized in Brazil until 1965 (*Schwartzman, 2001*). CNPq and CAPES remain the major funders of Brazilian academic science.

As the number of researchers increased, CAPES took up on the challenge of creating a national evaluation system for graduate education programs in Brazil (*Barata, 2016*). In the 1990s, the criteria for evaluation began to include quantitative indicators, such as numbers of articles published. In 1998, significant changes were made with the aim of trying to establish articles in international peer-reviewed journals as the main goal, and individual research areas were left free to design their own criteria for ranking journals. In 2007, amidst the largest-ever expansion in the number of federal universities, the journal ranking system in the life sciences became based on impact factors for the previous year, and remains so to this day (*CAPES, 2016*).

Today, Brazil has over 200,000 PhDs, with more than 10,000 graduating every year (*CGEE, 2016*). Although the evaluation system is seen as an achievement, it is subject to much criticism, revolving around the centralizing power of CAPES (*Hostins, 2006*) and the excessive focus on quantitative metrics (*Pinto and Andrade, 1999*). Many analysts criticize the country's research as largely composed of "salami science", growing in absolute numbers but lacking in impact, originality and influence (*Righetti, 2013*). Interestingly, research reproducibility has been a secondary concern in these criticisms, and awareness of the issue has begun to rise only recently.

With the economic and political crisis afflicting the country since 2014, science funding has suffered a sequence of severe cuts. As the Ministry for Science and Technology was merged with that of Communications, a recent constitutional amendment essentially froze science funding at 2016 levels for 20 years (*Angelo, 2016*). The federal budget for the Ministry suffered a 44% cut in 2017 and reached levels corresponding to roughly a third of those invested a decade earlier (*Floresti, 2017*), leading scientific societies to position themselves in defense of research funding (*SBPC, 2018*). Concurrently, CAPES has initiated discussions on how to reform its evaluation system (*ABC, 2018*). At this delicate moment, in which a new federal government has just taken office, an empirical assessment of the country's scientific output seems warranted to inform such debates.

## The Brazilian Reproducibility Initiative: aims and scope

The Brazilian Reproducibility Initiative was started in early 2018 as a systematic effort to evaluate the reproducibility of Brazilian biomedical science. Openly inspired by multicenter efforts such as the Reproducibility Project: Psychology (*Open Science Collaboration, 2015*), the Reproducibility Project: Cancer Biology (*Errington et al., 2014*) and the Many Labs projects (*Ebersole et al., 2016*; *Klein et al., 2014*; *Klein et al., 2018*), our goal is to replicate between 60 and 100 experiments from published Brazilian articles in the life sciences, focusing on common methods and performing each experiment in multiple sites within a network of collaborating laboratories. The project's coordinating team at the Federal University of Rio de Janeiro is responsible for the selection of methods and experiments, as well as for the recruitment and management of collaborating labs. Experiments are set to begin in mid-2019, in order for the project to achieve its final results by 2021.

Any project with the ambition of estimating the reproducibility of a country's science is inevitably limited in scope by the expertise of the participating teams. We will aim for the most representative sample that can be achieved without compromising feasibility, through the use of the strategies described below. Nevertheless, representativeness will be limited by the selected techniques and biological models, as well as by our inclusion and exclusion criteria – which include the cost and commercial availability of materials and the expertise of the replicating labs.

### Focus on individual experiments

Our first choice was to base our sample on experiments rather than articles. As studies in basic biomedical science usually involve many experiments with different methods revolving around a hypothesis, trying to reproduce a whole study, or even its main findings, can be cumbersome for a large-scale initiative. Partly because of this, the Reproducibility Project: Cancer Biology (RP:CB), which had originally planned to reproduce selected main findings from 50 studies, has been downsized to fewer than 20 (*Kaiser, 2018*). Moreover, in some cases RP:CB has been able to reproduce parts of a study but has also obtained results that cannot be interpreted or are not consistent with the original findings. Furthermore, the individual Replication Studies published by RP:CB do not say if a given replication attempt has been successful or not: rather, the project uses multiple measures to assess reproducibility.

Contrary to studies, experiments have well defined effect sizes, and although different criteria can be used for what constitutes a successful replication (*Goodman et al., 2016*; *Open Science Collaboration, 2015*), they can be defined objectively, allowing a quantitative assessment of reproducibility. Naturally, there is a downside in that replication of a single experiment is usually not enough to confirm or refute the conclusions of an article (*Camerer et al., 2018*). However, if one's focus is not on the studies themselves, but rather on evaluating reproducibility on a larger scale, we believe that experiments represent a more manageable unit than articles.

### Selection of methods

No replication initiative, no matter how large, can aim to reproduce every kind of experiment. Thus, our next choice was to limit our scope to common methodologies that are widely available in the country, in order to ensure that we will have a large enough network of potential collaborators. To provide a list of candidate methods, we started by performing an initial review of a sample of articles in Web of Science life sciences journals published in 2017, filtering for papers which: a) had all authors affiliated with a Brazilian institution; b) presented experimental results on a biological model; c) did not use clinical or ecological samples. One hundred randomly selected articles had data extracted concerning the models, experimental interventions and methods used to analyze outcomes:

the main results are shown in *Figure 1A and B*. A more detailed protocol for this step is available at https://osf.io/f2a6y/.

Based on this initial review, we restricted our scope to experiments using rodents and cell lines, which were by far the most prevalent models (present in 77 and 16% of articles, respectively). After a first round of automated full-text assessment of 5000 Brazilian articles between 1998 and 2017, we selected 10 commonly used techniques (*Figure 1C*) as candidates for replication experiments. An open call for collaborating labs within the country was then set up, and labs were allowed to register through an online form for performing experiments with one or more of these techniques and models during a three-month period. After this period, we used this input (as well as other criteria such as cost analysis) to select five methods for the replication effort: MTT assay, reverse transcriptase polymerase chain reaction (RT-PCR), elevated plus maze, western blot and immunohisto/cytochemistry (see https://osf.io/qxdjt/ for details). We are starting the project with the first three methods, while inclusion of the latter two will be confirmed after a more detailed cost analysis based on the fully developed protocols.

We are currently selecting articles using these techniques by full-text screening of a random sample of life sciences articles from the past 20 years in which most of the authors, including the corresponding one, are based in a Brazilian institution. From each of these articles, we select the first experiment using the technique of interest, defined as a quantitative comparison of a single outcome between two experimental groups. Although the final outcome of the experiment should be assessed using the method of interest, other laboratory techniques are likely to be involved in the model and experimental procedures that precede this step.

We will restrict our sample to experiments that: a) represent one of the main findings of the article, defined by mention of its results in the abstract; b) present significant differences between groups, in order to allow us to perform sample size calculations; c) use commercially available materials; d) have all experimental procedures falling within the expertise of at least three laboratories in our network; e) have an estimated cost below 0.5% of the project's total budget. For each included technique, 20 experiments will be selected, with the biological model and other features of the experiment left open to variation in order to maximize

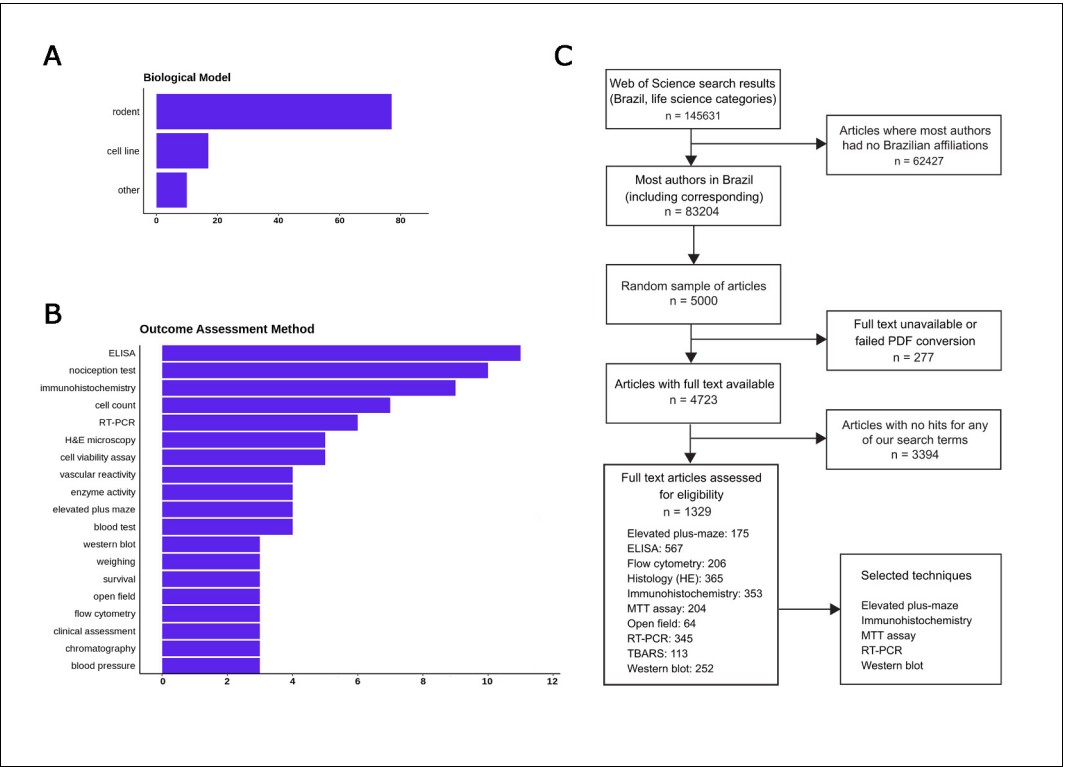

**Figure 1.** Selecting methods and papers for replication in the Brazilian Reproducibility Initiative. (**A**) Most frequent biological models used in main experiments within a sample of 100 Brazilian life sciences articles. (**B**) Most frequent methods used for quantitative outcome detection in these experiments. 'Cell count', 'enzyme activity' and 'blood tests' include various experiments for which methodologies vary and/or are not described fully in articles. Nociception tests, although frequent, were not considered for replication due to animal welfare considerations. (**C**) Flowchart describing the first full-text screening round to identify articles in our candidate techniques, which led us to select our final set of five methods.

DOI: https://doi.org/10.7554/eLife.41602.002

representativeness. A more detailed protocol for this step is available at https://osf.io/u5zdq/.

After experiments are selected, we will record each study's methods description in standardized description forms, which will be used to define replication protocols. These experiments will then be assigned to three laboratories each by the coordinating team, which will confirm that they have the necessary expertise in order to perform it.

### Multicenter replication

A central tenet of our project is that replication should be performed in multiple laboratories. As discussed in other replication projects (*Errington et al., 2014*; *Gilbert et al., 2016*; *Open Science Collaboration, 2015*) a single failed replication is not enough to refute the original finding, as there are many reasons that can explain discrepancies between results (*Goodman et al., 2016*). While some of them –

such as misconduct or bias in performing or analyzing the original experiment – are problematic, others – such as unrecognized methodological differences or chance – are not necessarily as alarming. Reproducibility estimates based on single replications cannot distinguish between these causes, and can thus be misleading in terms of their diagnoses (*Jamieson, 2018*).

This problem is made worse by the fact that data on inter-laboratory variability for most methods is scarce: even though simulations demonstrate that multicenter replications are an efficient way to improve reproducibility (*Voelkl et al., 2018*), they are exceedingly rare in most fields of basic biomedical science. Isolated attempts at investigating this issue in specific fields have shown that, even when different labs try to follow the same protocol, unrecognized methodological variables can still lead to a large amount of variation (*Crabbe et al., 1999*; *Hines et al., 2014*; *Massonnet et al., 2010*).

Thus, it might be unrealistic to expect that reproducing a published experiment – for which protocol details will probably be lacking (*Hair et al., 2018*; *Kilkenny et al., 2009*) – will yield similar results in a different laboratory.

In our view, the best way to differentiate irreproducibility due to bias or error from that induced by methodological variables alone is to perform replications at multiple sites. In this way, an estimate of inter-laboratory variation can be obtained for every experiment, allowing one to analyze whether the original result falls within the expected variation range. Multicenter approaches have been used successfully in the area of psychology (*Ebersole et al., 2016*; *Klein et al., 2014*; *Klein et al., 2018*), showing that some results are robust across populations, while others do not reproduce well in any of the replication sites.

Our plan for the Brazilian Reproducibility Initiative is to perform each individual replication in at least three different laboratories; this, however, opens up questions about how much standardization is desirable. Although one should follow the original protocol in a direct replication, there are myriad steps that will not be well described. And while some might seem like glaring omissions, such as the absence of species, sex and age information in animal studies (*Kilkenny et al., 2009*), others might simply be overlooked variables: for example, how often does one describe the exact duration and intensity of sample agitation (*Hines et al., 2014*)? When conditions are not specified, one is left with two choices. One of them is to standardize steps as much as possible, building a single, detailed replication protocol for all labs. However, this will reduce inter-laboratory variation to an artificially low level, making the original experiment likely to fall outside the effect range observed in the replications.

To avoid this, we will take a more naturalistic approach. Although details included in the original article will be followed explicitly in order for the replication to be as direct as possible, steps which are not described will be left open for each replication team to fill based on their best judgment. Replication teams will be required to record those choices in detailed methods description forms, but it is possible – and desirable – for them to vary according to each laboratory's experience. Methodological discrepancies in this case should approach those observed between research groups working independently, providing a realistic estimate of inter-laboratory variation for the assessment of published findings. This approach will also allow us to explore the impact of methodological variation on the experimental results – a topic perhaps as important as reproducibility itself – as a secondary outcome of the project.

### Protocol review

A central issue in other replication projects has been engagement with the original authors in order to revise protocols. While we feel this is a worthy endeavor, the rate of response to calls for sharing protocols, data or code is erratic (*Hardwicke and Ioannidis, 2018*; *Stodden et al., 2018*; *Wicherts et al., 2011*). Moreover, having access to unreported information is likely to overestimate the reproducibility of a finding based on published information, leading results to deviate from a 'naturalistic' estimate of reproducibility (*Coyne, 2016*). Thus, although we will contact the original authors for protocol details when these are available, in order to assess methodological variation between published studies and replications, this information will not be made available to the replication teams. They will receive only the protocol description from the published article, with no mention of its results or origin, in order to minimize bias. While we cannot be sure that this form of blinding will be effective, as experiments could be recognizable by scientists working in the same field, replicating labs will be stimulated not to seek this information.

Lastly, although non-described protocol steps will be left open to variation, methodological issues that are consensually recognized to reduce error and bias will be enforced. Thus, bias control measures such as blinding of researchers to experimental groups will be used whenever possible, and sample sizes will be calculated to provide each experiment with a power of 95% to detect the original difference – as in other surveys, we are setting our power estimates at a greater than usual rate due to the recognition that the original results are likely to be inflated by publication bias. Moreover, if additional positive and/or negative controls are judged to be necessary to interpret outcomes, they will also be added to the experiment.

To ensure that these steps are followed – as well as to adjudicate on any necessary protocol adaptations, such as substitutions in equipment or materials – each individual protocol will be reviewed after completion in a round-robin approach (*Silberzahn et al., 2018*) by (i) the project's coordinating team and (ii) an independent laboratory working with the same

technique that is not directly involved in the replication. Each of the three protocol versions of every experiment will be sent to a different reviewing lab, in order to minimize the risk of over-standardization. Suggestions and criticisms to the protocol will be sent back to the replicating team, and experiments will only start after both labs and the coordinating team reach consensus that the protocol: a) does not deviate excessively from the published one and can be considered a direct replication; b) includes measures to reduce bias and necessary controls to ensure the validity of results.

## Evaluating replications

As previous projects have shown, there are many ways to define a successful replication, all of which have caveats. Reproducibility of the general conclusions on the existence of an effect (e.g. two results finding a statistically significant difference in the same direction) might not be accompanied by reproducibility of the effect size; conversely, studies with effect sizes that are similar to each other might have different outcomes in significance tests (*Simonsohn, 2015*). Moreover, if non-replication occurs, it is hard to judge whether the original study or the replication is closer to the true result. Although one can argue that, if replications are conducted in an unbiased manner and have higher statistical power, they are more likely to be accurate, the possibility of undetected methodological differences preclude one from attributing non-replication to failures in the original studies.

Multisite replication is a useful way to circumvent some of these controversies, as if the variation between unbiased replications in different labs is known, it is possible to determine whether the original result is within this variability range. Thus, the primary outcome of our analysis will be the percentage of original studies with effect sizes falling within the 95% prediction interval of a meta-analysis of the three replications. Nevertheless, we acknowledge that this definition also has caveats: if inter-laboratory variability is high, prediction intervals can be wide, leading a large amount of results to be considered "reproducible". Thus, replication estimates obtained by these methods are likely to be optimistic. On the other hand, failed replications will be more likely to reflect true biases, errors or deficiencies in the original experiments (*Patil et al., 2016*).

An additional problem is that, given our naturalistic approach to reproducibility, incomplete reporting in the original study might increase inter-laboratory variation and artificially improve our primary outcome. With this in mind, we will include other ways to define reproducibility as secondary outcomes, such as the statistical significance of the pooled replication studies, the significance of the effect in a meta-analysis including the original result and replication attempts, and a statistical comparison between the pooled effect sizes of the replications and the original result. We will also examine thoroughness of methodological reporting as an independent outcome, in order to evaluate the possibility of bias caused by incomplete reporting.

Moreover, we will explore correlations between results and differences in particular steps of each technique; nevertheless, we cannot know in advance whether methodological variability will be sufficient to draw conclusions on these issues. As each experiment will be performed in only three labs, while there are myriad steps to each technique, it is unlikely that we will be able to pinpoint specific sources of variation between results of individual experiments. Nevertheless, by quantifying the variation across protocols for the whole experiment, as well as for large sections of it (model, experimental intervention, outcome detection), we can try to observe whether the degree of variation at each level correlates with variability in results. Such analyses, however, will only be planned once protocols are completed, so as to have a better idea of the range of variability across them.

Finally, we will try to identify factors in the original studies that can predict reproducibility, as such proxies could be highly useful to guide the evaluation of published science. These will include features shown to predict reproducibility in previous work, such as effect sizes, significance levels and subjective assessment by prediction markets (*Dreber et al., 2015*; *Camerer et al., 2016*; *Camerer et al., 2018*; *Open Science Collaboration, 2015*); the pool of researchers used for the latter, however, will be different from those performing replications, so as not to compromise blinding with respect to study source and results. Other factors to be investigated include: a) the presence of bias control measures in the original study, such as blinding and sample size calculations; b) the number of citations and impact factor of the journal; c) the experience of the study's principal investigator; d) the Brazilian region of origin; e) the technique used; f) the type of biological model; g) the area of research. As our sample of

experiments will be obtained randomly, we cannot ensure that there will be enough variability in all factors to explore them meaningfully. Nevertheless, we should be able to analyze some variables that have not been well explored in previous replication attempts, such as 'impact' defined by citations and publication venues, as most previous studies have focused on particular subsets of journals (*Camerer et al., 2018*; *Open Science Collaboration, 2015*) or impact tiers (*Errington et al., 2014*; *Ioannidis, 2005b*).

A question that cannot be answered directly by our study design is whether any correlations found in our sample of articles can be extrapolated either to different methods in Brazilian biomedical science or to other regions of the world. For some factors, including the reproducibility estimates themselves and their correlation with local variables, extrapolations to the international scenario are clearly not warranted. On the other hand, relationships between reproducibility and methodological variables, as well as with article features, can plausibly apply to other countries, although this can only be known for sure by performing studies in other regions.

All of our analyses will be preregistered at the Open Science Framework in advance of data collection. All our datasets will be made public and updated progressively as replications are performed – a process planned to go on until 2021. As an additional measure to promote transparency and engage the Brazilian scientific community in the project, we are posting our methods description forms for public consultation and review (see http://reprodutibilidade.bio.br/public-consultation), and will do so for the analysis plan as well.

## Potential challenges

A multicenter project involving the replication of experiments in multiple laboratories across a country of continental proportions is bound to meet challenges. The first of them is that the project is fully dependent on the interest of Brazilian laboratories to participate. Nevertheless, the response to our first call for collaborators exceeded our expectations, reaching a total of 71 laboratories in 43 institutions across 19 Brazilian states. The project received coverage by the Brazilian media (*Ciscati, 2018*; *Neves and Amaral, 2018*; *Pesquisa FAPESP, 2018*) and achieved good visibility in social networks, contributing to this widespread response. While we cannot be sure that all laboratories will remain in the project until its conclusion, it seems very

likely that we will have the means to perform our full set of replications, particularly as laboratories will be funded for their participation.

Concerns also arise from the perception that replicating other scientists' work indicates mistrust of the original results, a problem that is potentiated by the conflation of the reproducibility debate with that on research misconduct (*Jamieson, 2018*). Thus, from the start, we are taking steps to ensure that the project is viewed as we conceive it: a first-person initiative of the Brazilian scientific community to evaluate its own practices. We will also be impersonal in our choice of results to replicate, working with random samples and performing our analysis at the level of experiments; thus, even if a finding is not deemed reproducible, this will not necessarily invalidate an article's conclusions or call a researcher into question.

An additional challenge is to ensure that participating labs have sufficient expertise with a methodology or model to provide accurate results. Ensuring that the original protocol is indeed being followed is likely to require steps such as cell line/animal strain authentication and positive controls for experimental validation. Nevertheless, we prefer this naturalistic approach to the alternative of providing each laboratory with animals or samples from a single source, which would inevitably underestimate variability. Moreover, while making sure that a lab is capable of performing a given experiment adequately is a challenge we cannot address perfectly, this is a problem of science as a whole – and if our project can build expertise on how to perform minimal certification of academic laboratories, this could be useful for other purposes as well.

A final challenge will be to put the results into perspective once they are obtained. Based on the results of previous reproducibility projects, a degree of irreproducibility is expected and may raise concerns about Brazilian science, as there will be no estimates from other countries for comparison. Nevertheless, our view is that, no matter the results, they are bound to put Brazil at the vanguard of the reproducibility debate, if only because we will likely be the first country to produce such an estimate.

## Conclusions

With the rise in awareness over reproducibility issues, systematic replication initiatives have begun to develop in various research fields (*Camerer et al., 2016*; *Camerer et al., 2018*;

*Cova et al., 2018*; *Errington et al., 2014*; *Open Science Collaboration, 2015*; *Tan et al., 2015*). Our study offers a different perspective on the concept, covering different research areas in the life sciences with focus in a particular country.

This kind of initiative inevitably causes controversy both on the validity of the effort (*Coyne, 2016*; *Nature Medicine, 2016*) and on the interpretation of the results (*Baker and Dolgin, 2017*; *Gilbert et al., 2016*; *Patil et al., 2016*). Nevertheless, multicenter replication efforts are as much about the process as about the data. Thus, if we attain enough visibility within the Brazilian scientific community, a large part of our mission – fostering the debate on reproducibility and how to evaluate it – will have been achieved. Moreover, it is healthy for scientists to be reminded that self-correction and confirmation are a part of science, and that published findings are passive of independent replication. There is still much work to be done in order for replication results to be incorporated into research assessment (*Ioannidis, 2014*; *Munafò et al., 2017*), but this kind of reminder by itself might conceivably be enough to initiate cultural and behavioral change.

Finally, for those involved as collaborators, one of the main returns will be the experience of tackling a large scientific question collectively in a transparent and rigorous way. We believe that large-scale efforts can help to lead an overly competitive culture back to the Mertonian ideal of communality, and hope to engage both collaborators and the Brazilian scientific community at large through data sharing, public consultations and social media (via our website: http://reprodutibilidade.bio.br/home). The life sciences community in Brazil is large enough to need this kind of challenge, but perhaps still small enough to answer cohesively. We thus hope that the Brazilian Reproducibility Initiative, through its process as much as through its results, can have a positive impact on the scientific culture of our country for years to come.

**Olavo B Amaral** is in the Institute of Medical Biochemistry Leopoldo de Meis, Federal University of Rio de Janeiro, Rio de Janeiro, Brazil

olavo@bioqmed.ufrj.br

https://orcid.org/0000-0002-4299-8978

**Kleber Neves** is in the Institute of Medical Biochemistry Leopoldo de Meis, Federal University of Rio de Janeiro, Rio de Janeiro, Brazil

https://orcid.org/0000-0001-9519-4909

**Ana P Wasilewska-Sampaio** is in the Institute of Medical Biochemistry Leopoldo de Meis, Federal University of Rio de Janeiro, Rio de Janeiro, Brazil

https://orcid.org/0000-0003-0378-3883

**Clarissa FD Carneiro** is in the Institute of Medical Biochemistry Leopoldo de Meis, Federal University of Rio de Janeiro, Rio de Janeiro, Brazil

https://orcid.org/0000-0001-8127-0034

*Author contributions:* Olavo B Amaral, Conceptualization, Supervision, Funding acquisition, Methodology, Writing—original draft, Project administration, Writing—review and editing; Kleber Neves, Data curation, Software, Formal analysis, Investigation, Visualization, Methodology, Writing—review and editing; Ana P Wasilewska-Sampaio, Data curation, Investigation, Visualization, Methodology, Project administration, Writing—review and editing; Clarissa FD Carneiro, Data curation, Formal analysis, Supervision, Investigation, Methodology, Writing—review and editing

*Competing interests:* The authors declare that no competing interests exist.

### Funding

| Funder | Author |
| --- | --- |
| Instituto Serrapilheira | Olavo B Amaral |
| Conselho Nacional de Desenvolvimento Científico e Tecnológico | Clarissa FD Carneiro |

The project's funder (Instituto Serrapilheira) made suggestions on the study design, but had no role in data collection and interpretation, or in the decision to submit the work for publication. KN and APWS are supported by post-doctoral scholarships within this project. CFDC is supported by a PhD scholarship from CNPq.

**Decision letter and Author response**
Decision letter https://doi.org/10.7554/eLife.41602.008
Author response https://doi.org/10.7554/eLife.41602.009

## Additional files
### Supplementary files
• Transparent reporting form
DOI: https://doi.org/10.7554/eLife.41602.003

### Data availability
All data cited in the article is available at the project's site at the Open Science Framework (https://osf.io/6av7k/).

The following dataset was generated:

| Author(s) | Year | Dataset URL | Database and Identifier |
|---|---|---|---|
| Amaral OB, Neves K, Wasilewska-Sampaio AP, Carneiro CFD | 2018 | https://osf.io/6av7k/ | Open Science Framework, 10.17605/OSF.IO/6AV7K |

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
