## [Decision Letter]

Thank you for submitting your article "The Brazilian Reproducibility Initiative: a systematic assessment of Brazilian biomedical science" to *eLife* for consideration as a Feature Article. Your article has been reviewed by two peer reviewers, and the evaluation has been overseen by the Peter Rodgers, *eLife* Features Editor. The following individuals involved in review of your submission have agreed to reveal their identity: Timothy M Errington (Reviewer #1); Richard Klein (Reviewer #2).

The reviewers have discussed the reviews with one another and the Reviewing Editor has drafted this decision to help you prepare a revised submission.

Summary:

There is a clear need for these sorts of projects to re-evaluate where we stand, across all scientific disciplines, in light of the replication crises already observed in psychology and other fields. This project seems well organized and thought-out, but could be strengthened by addressing the following comments.

Essential revisions:

a) The article needs to be as up-to-date as possible when published. For example, if the 3-5 methods have been identified, the article needs to say what they are and how they were chosen; if they have not been identified, the description of the selection process needs to be improved (see point j below). Likewise for the recruitment of collaborators.

b) I agree that narrowing the replications to a signal experiment (really a single main effect) is a clean way to measure reproducibility across studies. However, there are many 'nested' aspects to an experiment that need to be considered for proper interpretation. For example, testing the impact of depleting a particular gene of interest on some outcome (e.g. apoptosis) could require multiple experimental techniques (e.g. PCR or Western blot to confirm depletion, flow cytometry to detect apoptosis). Will all control related experimental aspects be included in the replication (in the example above PCR or Western blot) in addition to the key outcome (flow cytometry)? Also, it sounds as if just a single comparison (e.g. treatment vs control; https://osf.io/57f8s/) will be performed for each experiment: will positive and/or negative controls also be included?

c) Subsection “The Brazilian Reproducibility Initiative: aims and scope”, second paragraph. Please be more explicit about how generalizable your results will be to biomedical research in Brazil, Brazilian science in general, and biomedical research in general, given your sample size and biases imposed by your inclusion/exclusion criteria (e.g. use of commercially available materials (subsection “Selection of methods”, last paragraph).

d) Please also be more explicit about the caveats associated with having only a few labs if one goal of the project is to identify particular steps of a technique that are associated with a lack of reproducibility. There are numerous steps in a given technique and little variation with only 3 replications (as mentioned in the third paragraph of the subsection “Evaluating replications”). Nonetheless, the added benefit on the outcome by allowing for variation to occur will give a better understanding of the reproducibility (generalizability) of the finding across settings.

e) How much variation will be allowed? For example, if the original study states the species or sex of an animal, will that be held constant or will it be open to change among the labs? If a specific drug is stated, will that be kept constant or will it be open to change if it has the same MOA? Each lab should preregister experiments based on a common framework of what will not be changed (i.e. what they and the project leads agree are perceived necessary features of the original experiment). The worry I have is that if left open to interpretation too broadly, the three experiments that are replications will be more conceptual than direct and less comparable to each other or the original as much as the authors intend (see: Nosek and Errington, 2017; doi.org/10.7554/*eLife*.23383).

f) Two possible questions:

1) Can the findings be reproduced based on the published literature?

2) Can the findings be reproduced under *ideal* conditions?

Both are extremely important, but by blinding teams to feedback/contact with original authors you *may* be favoring #1 at some expense to #2. You might consider (but do not have to do this) experimentally testing for the influence of original author feedback by giving one of the three labs for each site any extra details provided by the original authors (and perhaps further ask them to review that lab's procedure). This could then both estimate reproducibility and inform which solutions to prioritize to improve it (e.g., improved documentation). No idea what response rate you'd expect, but in Many Labs 2 we got feedback from original authors or close collaborators in 28/28 studies.

g) I appreciate the difficulty defining replication success, but if the most important outcome is whether an effect is observed or not then I'm not sure the current primary definition is optimal. You may consider defining a 'smallest effect size of interest' and combining that with significance testing (e.g., r >.1 and p <.001).

h) It is not clear how the individual studies and the overall paper will be peer reviewed. Will the individual studies be handled in a similar way to the RP:CB? (That is, there is a Registered Report for each individual study that is reviewed and must be accepted before data collection can begin, and there is an agreement to publish the results, regardless of outcome, so long as the protocol in the Registered Report has been adhered to). And what process will be used to peer review the overall paper? I highly recommend a structured procedure to ensure quality and combat any possible publication bias against null findings. For most Many Lab projects we've included original authors as part of this review process, but I understand the arguments against this.

i) Please include a flow chart (such as a PRISMA flow diagram) to show how the papers to be replicated will be selected. I also have some questions about the selection process as described in the present manuscript. The text states that 5000 articles were assessed (subsection “Selection of methods”, second paragraph), but the table legend mentions an initial survey of 100 articles (is that 100 of the 5000?). Also, if I count the number of occurrences of each technique among the main experiments, I count 51, not 100, which suggests 49 are being excluded for some reason. Please also consider including a figure along the lines of Figures 3-5 in https://osf.io/qhyae/ to show the range of techniques used.

j) Subsection “Selection of methods”, second paragraph: How will the list of 10 commonly used techniques (shown in Table 1) be narrowed down to 3-5 methods? Will cost be a factor? Also, will the distribution of biological models used in the replications match the overall distribution (i.e., rodents > cell lines > micro-organisms etc.), or will simple randomization occur? And likewise, for the techniques used?

---

## [Author Response]

Essential revisions:a) The article needs to be as up-to-date as possible when published. For example, if the 3-5 methods have been identified, the article needs to say what they are and how they were chosen; if they have not been identified, the description of the selection process needs to be improved (see point j below). Likewise for the recruitment of collaborators.

We naturally agree with the point of updating the manuscript with the current state of the project. In this sense, the revision process has allowed some important steps to be concluded. Over 3 months of registration, we’ve had 71 laboratories across 19 states of Brazil sign up as potential collaborators of the initiative.

On the basis of this network (as well as of methodological and budget concerns, as will be described below), we have selected five methods to be included in the replication effort: namely, MTT, RT-PCR, elevated plus maze, western blot and immunohistochemistry. We will start the replication experiments with three of these (MTT, RT-PCR and elevated plus maze), totaling 60 experiments, and will add the other two after full protocol development for the first three, as this will allow us to have a clearer estimation of the project’s workload and costs.

These details have now been updated in the third paragraph of the subsection “Selection of methods”, and in other points of the manuscript. The selection process has also been described in clearer detail in the flowchart presented in Figure 1C, as suggested below, which has replaced Table 1 of the original manuscript.

b) I agree that narrowing the replications to a signal experiment (really a single main effect) is a clean way to measure reproducibility across studies. However, there are many 'nested' aspects to an experiment that need to be considered for proper interpretation. For example, testing the impact of depleting a particular gene of interest on some outcome (e.g. apoptosis) could require multiple experimental techniques (e.g. PCR or Western blot to confirm depletion, flow cytometry to detect apoptosis). Will all control related experimental aspects be included in the replication (in the example above PCR or Western blot) in addition to the key outcome (flow cytometry)? Also, it sounds as if just a single comparison (e.g. treatment vs control; https://osf.io/57f8s/) will be performed for each experiment: will positive and/or negative controls also be included?

The review points out an important point that we agree was not completely clear in the original manuscript. Our article/experiment selection will be based on the main technique of interest – i.e. that used to measure the outcome variable. That said, as the whole experiment will be replicated, there are naturally other methods that will be involved in performing the required experimental interventions, and eventually for adding necessary controls.

In our recruitment process, we asked for reasonably detailed information on the expertise of the participating laboratories – not only in the main techniques of the project, but also in handling biological models, performing interventions, dissecting tissue, etc. Thus, one of the criteria for inclusion of articles will be that we have at least three participating labs with the required expertise to perform the whole experiment. This will be confirmed with the replicating labs after screening to confirm inclusion of the experiment. If a given experiment requires expertise that is not available in at least three labs, it will not be included in the replication initiative. This is now explained in the fourth and last paragraphs of the subsection “Selection of methods”.

Our main result of interest will be indeed based on a single experiment – i.e. a comparison in a dependent variable between two groups – in order to facilitate statistical analysis for the whole sample. Nevertheless, the reviewers are right in pointing out that, in some experiments, additional controls might be necessary for interpreting the results – e.g. positive controls to confirm the sensitivity of the detection method, for example. If such controls are part of the original experiment, they will be included in the replication as well, unless for some reason this is not technically feasible. If not, the replicating teams will still be allowed to suggest controls when they are judged necessary. The need for these controls will be reviewed during the protocol revision process – see response to point (h) below – in order to confirm inclusion, as explained in the seventh paragraph of the subsection “Multicenter replication”.

c) Subsection “The Brazilian Reproducibility Initiative: aims and scope”, second paragraph. Please be more explicit about how generalizable your results will be to biomedical research in Brazil, Brazilian science in general, and biomedical research in general, given your sample size and biases imposed by your inclusion/exclusion criteria (e.g. use of commercially available materials (subsection “Selection of methods”, last paragraph).

Generalizability of the results will certainly be limited by many factors, including the selected techniques/biological models and our inclusion/exclusion criteria for selecting experiments – which include cost, commercial availability and expertise of the replicating labs. This is now made clear not only in the above-mentioned passage (subsection “The Brazilian Reproducibility Initiative: aims and scope”, second paragraph) but also in the sixth paragraph of the subsection “Evaluating replications”.

d) Please also be more explicit about the caveats associated with having only a few labs if one goal of the project is to identify particular steps of a technique that are associated with a lack of reproducibility. There are numerous steps in a given technique and little variation with only 3 replications (as mentioned in the third paragraph of the subsection “Evaluating replications”). Nonetheless, the added benefit on the outcome by allowing for variation to occur will give a better understanding of the reproducibility (generalizability) of the finding across settings.

The reviewers are right in pointing out that our ability to detect variation due to any particular steps of the selected techniques will be limited by the number of replicating labs and by the amount of variation between protocols – which cannot be predicted in advance, as it will depend on how different labs will interpret and adapt the published protocols. Nevertheless, as our primary goal is to examine the reproducibility of the published literature in view of naturalistic interlaboratory variability, this is a necessary compromise: for laboratories to be free in making their own choices, we must refrain from controlling variation.

In view of this, we are fully aware both that (a) investigation of the effect of individual steps on variability will be a secondary analysis which will be limited in terms of scope and statistical power and that (b) it is hard to predict in advance how limited this scope will be. For this reason, we will work on the analysis plan for this part of the project only after protocols have been built (but before the experiments are performed), in order to have a better idea of the range of methodological variability. More likely than not, we will have to substitute analysis of individual steps of the methods for large-scale, quantitative estimates of variability across the whole experiment or its general sections.

That said, we point out that, although each experiment will be performed only in three labs, we will have 20 experiments with each technique with the same methodological steps described. Thus, even if in a single experiment it may be impossible to pinpoint sorts of variation, on the aggregate we might have some idea on what sources of methodological variability correlate more strongly with variation in results.

An in-depth discussion of these analysis options seems premature at this point of the project – and is certainly beyond the scope of the current manuscript. Nevertheless, we now acknowledge the above-mentioned limitations more clearly in the fourth paragraph of the subsection “Evaluating replications”.

e) How much variation will be allowed? For example, if the original study states the species or sex of an animal, will that be held constant or will it be open to change among the labs? If a specific drug is stated, will that be kept constant or will it be open to change if it has the same MOA? Each lab should preregister experiments based on a common framework of what will not be changed (i.e. what they and the project leads agree are perceived necessary features of the original experiment). The worry I have is that if left open to interpretation too broadly, the three experiments that are replications will be more conceptual than direct and less comparable to each other or the original as much as the authors intend (see: Nosek and Errington, 2017; doi.org/10.7554/eLife.23383).

As stated in the manuscript (subsection “Multicenter replication”, fifth paragraph), replications will be as direct as possible in the sense of explicitly following the protocol described in the original manuscript. Naturally, there will be cases in which adaptations are required – e.g. use of different equipment or reagent when the original is not available. Nevertheless, all these adaptations will be revised by the coordinating team and an independent laboratory (see protocol review process below) so that it is agreed that the adapted protocol maintains the necessary features of the original experiment.

On the other hand, steps that are not described in the original protocol will be left to vary as much as possible. This is necessary to approach our main goal of measuring naturalistic reproducibility – i.e. question 1 in point (f) below. Although these steps will be left to vary, we will record these protocol variations in as much details as possible using standardized forms, in order to allow for the analyses mentioned in the point above.

An exception to the rule of following the original protocol as closely as possible will be if we feel that additional steps are necessary to ensure that the experiments are performed free of bias – for example, unblinded experiments will be blinded and non-randomized experiments will be randomized whenever possible. As described above, we will also add necessary controls when deemed necessary (even though the main experiment is left unchanged). This is now explained in more detail in the seventh paragraph of the subsection “Multicenter replication”.

f) Two possible questions:1) Can the findings be reproduced based on the published literature?2) Can the findings be reproduced under ideal conditions?Both are extremely important, but by blinding teams to feedback/contact with original authors you may be favoring #1 at some expense to #2. You might consider (but do not have to do this) experimentally testing for the influence of original author feedback by giving one of the three labs for each site any extra details provided by the original authors (and perhaps further ask them to review that lab's procedure). This could then both estimate reproducibility and inform which solutions to prioritize to improve it (e.g., improved documentation). No idea what response rate you'd expect, but in Many Labs 2 we got feedback from original authors or close collaborators in 28/28 studies.

The reviewers are correct in stating that these two questions are different – and perhaps mutually exclusive. Our option from the start of the project has always been to answer question (1) – i.e. reproducibility in naturalistic conditions upon reading the literature, and we are indeed favoring it deliberately. This has been made clear in the fifth paragraph of the subsection “Multicenter replication”.

The suggestion to try to address both questions at the same time is certainly interesting – however, we feel that it could complicate the analysis of the results. As some replications will be more alike to the original experiment than the others, our estimate of interlaboratory variability would be slightly more biased towards the original experiment than what would be expected in naturalistic replication. We can solve this by removing these labs from the analysis, but that would leave only 2 laboratories performing truly independent, unguided replications.

Another problem with this approach is that, even if we do get feedback from most original authors – something on which we are not clearly counting on – we will be replicating experiments published over 20 years (e.g. between 1998 and 2017), which means that much protocol information will likely have been lost for older experiments. Moreover, responder bias among authors could also cause information to be more available for particular types of authors or institutions; thus, using author information only when it’s available would likely bias our analyses concerning predictors of reproducibility.

With this in mind, although we find the suggestion interesting, we would rather keep up with our original plot of focusing on naturalistic reproducibility based only on published information, as explained inthe sixth paragraph of the subsection “Multicenter replication”.

g) I appreciate the difficulty defining replication success, but if the most important outcome is whether an effect is observed or not then I'm not sure the current primary definition is optimal. You may consider defining a 'smallest effect size of interest' and combining that with significance testing (e.g., r >.1 and p <.001).

We also appreciate the difficulty in defining replication success – and have opted for multiple measures, as previous replication initiatives have done, precisely because of that. That said, we are somewhat skeptical of defining a ‘smallest effect size of interest’, as ‘typical’ effect sizes seem to vary a lot across areas of science depending on the type of interventions that are used. Our own work in rodent fear conditioning, for example (Carneiro et al., 2018) shows that the arbitrary criteria for ‘small’, ‘moderate’ and ‘large’ effects commonly used in psychology do not hold at all in that context. Moreover, the importance of effect sizes is unfortunately neglected in most fields of biology (again, see Carneiro et al., 2018 for an example). Thus, as we will be looking across different areas of science and interventions, we do not feel confident in defining a smallest effect size of interest.

The statistical significance of the pooled replication studies is already one of our definitions of a successful replication, as has been the case in most large replication initiatives. Nevertheless, we would prefer not to be so stringent as to lower the significance threshold to p < 0.001. In keeping in line with other replication studies, and with most of the biomedical literature, we would prefer to keep up with the standard p < 0.05. We fully understand that this means that some false positives will be expected within the replication effort, but we feel maintaining the declared false-positive rate of the original studies provides a fairer assessment of whether these studies might have contained unjustified claims. Nevertheless, we commit to reporting exact p values for all experiments, in order to allow our data to be interpreted under different threshold assumptions.

h) It is not clear how the individual studies and the overall paper will be peer reviewed. Will the individual studies be handled in a similar way to the RP:CB? (That is, there is a Registered Report for each individual study that is reviewed and must be accepted before data collection can begin, and there is an agreement to publish the results, regardless of outcome, so long as the protocol in the Registered Report has been adhered to). And what process will be used to peer review the overall paper? I highly recommend a structured procedure to ensure quality and combat any possible publication bias against null findings. For most Many Lab projects we've included original authors as part of this review process, but I understand the arguments against this.

We thank the reviewers for bringing the point about peer review of the protocols, which we had not addressed in the original manuscript – partly because we had not discussed it in sufficient depth at the time of submission. We certainly do not plan to publish each experiment as a peer-reviewed Registered Report, as the individual experiments by themselves (unlike the larger studies included in RP:CB) would probably be of little interest to readers. That said, we agree on the importance of peer review happening before experiments are performed.

For this, we will use a similar approach to that used in the recent “Many Analysts, One Dataset” project (Silberzahn et al., 2018) and use our own base of collaborating labs – which we now know to be large enough – to provide peer review on the first version of the protocols. Each protocol will be reviewed both by a member of the coordinating team and by an independent laboratory participating in the Initiative with expertise in the same technique. These reviewers will be instructed to (a) verify whether there are adaptations to the originally described protocol and adjudicate whether any of them constitute a deviation that is large enough to make it invalid as a direct replication and (b) verify whether necessary measures and controls to reduce risk of bias and ensure the validity of results were taken. After peer review, each laboratory will receive feedback from both reviewers and will be allowed to revise the protocol, which will be reviewed once more by the same reviewers for approval before the start of the experiments. This is now discussed in the last paragraph of the subsection “Multicenter replication”.

We will review each protocol independently – i.e. each of the 3 different versions of an experiment will be reviewed by a different reviewer – in order to minimize the possibility of ‘overstandardization’ – i.e. to prevent that having a single reviewer revising all the protocols would make them more similar than would be expected from independent replication. In keeping up with the idea of naturalistic replication, as detailed above – as well as to preserve blinding – we will not include feedback from the original authors as part of the review – although we also understand the argument in favor of this. We discuss all of these issues in the aforementioned subsection.

Concerning the publication plan itself, our current view is that the actual article describing the whole set of results will be submitted as a regular paper after completion of the experiments – even though all protocols will have been reviewed and preregistered at the Open Science Framework in advance. We also envision that the results within each technique might deserve a separate, in-depth article examining sources of variability within each kind of experiment and their relationship to reproducibility. Finally, it is possible that some of the secondary analyses – such as researcher predictions and correlates of reproducibility, could generate other articles, although we would rather keep these in the main publication. Such concerns, however, are probably premature at the moment, and the ideal form of publication will become clearer as the project develops.

i) Please include a flow chart (such as a PRISMA flow diagram) to show how the papers to be replicated will be selected. I also have some questions about the selection process as described in the present manuscript. The text states that 5000 articles were assessed (subsection “Selection of methods”, second paragraph), but the table legend mentions an initial survey of 100 articles (is that 100 of the 5000?). Also, if I count the number of occurrences of each technique among the main experiments, I count 51, not 100, which suggests 49 are being excluded for some reason. Please also consider including a figure along the lines of Figures 3-5 in https://osf.io/qhyae/ to show the range of techniques used.

We thank the reviewer for the excellent suggestion concerning the flow chart for selection of experiments. It has now been included as Figure 1C. That said, experiment selection will continue over the next months, with the groups of participating labs already defined, in order to arrive at a sample of experiments that is within our available expertise.

The initial survey was performed separately from the article selection process, as in the former our goal was to assess candidate techniques widely performed within the country. For this reason, in this step we only worked with articles in which all authors were located in Brazil, in order to make sure that the methods were available locally. The main results of the survey are now included in Figure 1A and 1B. A wide range of methods was detected in this step, and the figure only shows the most common results; thus, the results shown in the bar graphs are non-extensive and do not add up to 100.

j) Subsection “Selection of methods”, second paragraph: How will the list of 10 commonly used techniques (shown in Table 1) be narrowed down to 3-5 methods? Will cost be a factor? Also, will the distribution of biological models used in the replications match the overall distribution (i.e., rodents > cell lines > micro-organisms etc.), or will simple randomization occur? And likewise, for the techniques used?

Our list of candidate techniques has now been narrowed down to 5, as explained in the third paragraph of the subsection “Selection of methods” and in Figure 1C. The criteria used to define these techniques were (a) the number of labs with expertise in this methods that signed up for the initiative, (b) the availability of experiments in an initial screening sample that matched the labs’ expertise while remaining under our cost limit and (c) the total number of labs that could be included with the particular combination chosen. Thus, even though fewer laboratories signed up as candidates for behavioral techniques, the elevated plus maze was included due to its low cost and to the availability of a pool of laboratories that was largely independent from those signed up for the molecular techniques (e.g. MTT, RT-PCR, western blot).

Concerning the number of experiments included, our aim is to include 20 experiments per technique – thus, the distribution of techniques is set a priori and will not match their actual prevalence within the literature. Within each technique, however, the biological models will not be constrained – thus, the prevalence of experiments using rodents and cell lines should approach that observed in the literature for each method, although this prevalence could be biased by cost or expertise issues. This is now better explained in the fifth paragraph of the subsection “Selection of methods”.